# Primitive noble gases sampled from ocean island basalts cannot be from the Earth's core

Yunguo Li [1,2,3 ✉], Lidunka Vočadlo[1], Chris Ballentine [4] & John P. Brodholt [1,5]

Noble gas isotopes in plumes require a source of primitive volatiles largely isolated in the Earth for 4.5 Gyrs. Among the proposed reservoirs, the core is gaining interest in the absence of robust geochemical and geophysical evidence for a mantle source. This is supported by partitioning data showing that sufficient He and Ne could have been incorporated into the core to source plumes today. Here we perform ab initio calculations on the partitioning of He, Ne, Ar, Kr and Xe between liquid iron and silicate melt under core forming conditions. For He our results are consistent with previous studies allowing for substantial amounts of He in the core. In contrast, the partition coefficient for Ne is three orders of magnitude lower than He. This very low partition coefficient would result in a $^{3}He/^{22}Ne$ ratio of ~$10^3$ in the core, far higher than observed in ocean island basalts (OIBs). We conclude that the core is not the source of noble gases in OIBs.

[1] Department of Earth Sciences, University College London, London, UK. [2] CAS Key Laboratory of Crust-Mantle Materials and Environments, School of Earth and Space Sciences, University of Science and Technology of China, Hefei, China. [3] CAS Center for Excellence in Comparative Planetology, School of Earth and Space Sciences, University of Science and Technology of China, Hefei, China. [4] Department of Earth Sciences, University of Oxford, Oxford, UK. [5] Centre for Earth Evolution and Dynamics, University of Oslo, Oslo, Norway. ✉email: liyunguo@ustc.edu.cn

Inert noble gases are important geodynamic tracers that can be used to decode the Earth's accretion, convection and degassing processes. The high non-radiogenic to radiogenic noble gas isotope ratios in ocean island basalts (OIBs) have been interpreted as evidence of a primordial volatile reservoir in the deep Earth[1]. In particular, the ratios of primordial $^3$He to $^4$He in OIBs range from 5 to 50 $Ra$[2–4] ($Ra$ is the $^3$He/$^4$He of modern air) while those in mid-ocean-ridge basalts (MORBs) range from 6 to 12 $Ra$[2]. Although not all OIBs have high $^3$He/$^4$He, the highest values (>15) are always associated with OIBs. OIBs with high $^3$He/$^4$He are often accompanied by other characteristic geochemical tracers including $^3$He/$^{22}$Ne[1], $^{129}$Xe/$^{130}$Xe[5], $^{40}$Ar/$^{36}$Ar[6] and $^{20}$Ne/$^{22}$Ne[7], consistent with the existence of a volatile reservoir preserving primordial geochemical signals despite mantle convection over the last 4.5 Gyrs.

Different reservoirs have been proposed for the source of primordial gases sampled by plumes. Initial ideas suggesting that plumes sample the lower mantle while MORs only sample the upper mantle fell out of favour when seismic tomography showed slabs subducting into the lower mantle[8] and that mantle mixing is likely to be efficient[9]. Since then, Large Low Shear Velocity Provinces (LLSVPs)[10,11], Ultra-low-velocity zones (ULVZs) and Bridgmanite-enriched regions of the mantle (BEAMS) have all been suggested as potential mantle hosts; however, none are without problems. For instance, while some plumes with high $^3$He/$^4$He anomalies seem to be associated with LLSVPs[11], the connection between plumes and the LLSVPs remains unclear[12]. Moreover, some plumes associated with LLSVPs have Nd isotopic signatures suggesting that at least some part of LLSVPs are made of recycled oceanic crustal material[12], consistent with the recently measured low velocities of MORB[13,14]. BEAMs have been suggested to be very early crystallisation products of the magma ocean, but it is not clear how they would obtain, retain and then later release incompatible noble gases. And unlike LLSVPs and ULVZs, there is no seismic evidence for BEAMs. ULVZs are small dense bodies on the core-mantle boundary but little else is known about them. They are likely to be Fe-rich[15] and may or may not be partially molten[16]. While some ULVZs are located near to, or within, LLSVPs, their link to the location of surface hotspots is not obvious[17].

An alternative to a silicate reservoir is the Earth's core[18–20]. The core is an attractive option since it has unequivocally been isolated from the convective mantle for the age of the Earth. It is also large and so even low concentrations of noble gases would influence mantle reservoirs if able to diffuse out. It had originally been thought that the concentration of noble gases partitioned into the core, including He, are insufficient to sustain such a reservoir[21], but more recent experiments found a change in partitioning behaviour with pressure showing that sufficient He and Ne can partition into the core during its formation[20]. However, this work is based on the low-pressure data (<16 GPa and 3000 K) of recovered samples of liquid iron and silicate melt from experiments and a two-phase ab initio molecular dynamics (AIMD) simulation at 40 GPa[22], whereas actual core-mantle differentiation conditions could be as high as ~100 GPa and 3000–4500 K[23]. Moreover, partitioning data for the other noble gases is currently unavailable. Core-mantle differentiation is highly likely to be a polybaric process as metal droplets will re-equilibrate progressively on the way to the core. Indeed, Rubie et al.[23] modelled the Earth's accretion and differentiation by considering the partitioning of Fe, Si, O, Ni, Co, W, Nb, V, Ta and Cr, and found an average equilibration at pressures larger than 50 GPa is required for core formation. This average high-pressure differentiation is also consistent with the highly siderophile elements abundances in the mantle[24]. Using the existing low-pressure data to constrain the partitioning of the core is, therefore, likely to have large uncertainties.

To test the hypothesis of a core reservoir, we use ab initio thermodynamic techniques to obtain the chemical potential of the noble gases (He, Ne, Ar, Kr and Xe, denoted as X) in both silicate melts and Fe melts (see Methods). These are then used to derive the partition coefficients of noble gases between liquid iron (l-Fe) and silicate melt (l-MgSiO$_3$) at 50 GPa (3500 K) and 135 GPa (4200 K). Using these partitioning coefficients, we then estimate the concentration of noble gases in the core and compare their isotopic ratios and elemental ratios to those in hotspots to evaluate if the core can be the long sought-after host of noble gases.

## Results and discussion

**Partition coefficients.** We first focus on He partition coefficients, which are shown together with the available literature data in Fig. 1a. As can be seen, our 20 GPa partition coefficient lies within the range of experimental data and and agrees with the calculation by Xiong et al.[25]. Our 50 GPa partition coefficient is in good agreement with the highest pressure experimental data and the calculations of Zhang and Yin[22] At higher pressure we predict $D_{\text{He}}^{l-\text{Fe}/l-\text{MgSiO}_3}$ (all partition coefficients discussed here are for liquid iron and silicate melt, unless stated otherwise) to be somewhat lower than the ab initio predictions of Xiong et al.[25] As discussed in detail in the Supplementary Information, Xiong et al. calculate the partitioning based on the free energies using only one concentration of He in each phase rather than the five or six used here, and so will have much larger uncertainties in D of about 3 orders of magnitude. Nevertheless, while less accurate, their results are in general agreement with ours.

As can be seen in Fig. 1a, He partitioning between silicate melt and Fe melt seems to have a complex behaviour with pressure, although some of it may be due to uncertainties in experimental and theoretical estimates. For instance, At the lowest pressures, the experiments of Matsuda show an apparent drop in $D_{\text{He}}^{l-\text{Fe}/l-\text{MgSiO}_3}$ of about an order of magnitude below 2 and 5 GPa, however this is not replicated in the later experiments of Bouhifd et al. Between about 5 and 40 GPa, $D_{\text{He}}^{l-\text{Fe}/l-\text{MgSiO}_3}$ increases, by about an order of magnitude, after which our results predict it to decrease again with pressure. Such a trend can be correlated with the evolution of the silicate-melt structure since silicate melts exhibit strong structural changes at pressures up to ~40 GPa[26] and, in particular, the average coordination number of oxygens bonded to Mg increases from about 5 to 8. Since noble gases tend to act like oxygen and are mostly coordinated to Mg (see discussion below), the change in coordination is likely to affect the partitioning. After ~40 GPa structural changes become less pronounced[26–28] and partitioning is likely to be controlled by bond compression. Xiong et al.'s results show a general increases in D with depth, however, this may be due to much greater uncertainties due to the free energy on being calculated at a single concentration of noble gas. Either way, all results are consistent with $D_{\text{He}}^{l-\text{Fe}/l-\text{MgSiO}_3}$ being between $10^{-2}$ and $10^{-3}$ over all lower mantle conditions. We have not considered the newer results of Wang et al.[29] since they are also based on an incorrect use of the free energy in obtaining Ds. As we show in the Supplementary Information, when using their Gs to equate chemical potentials, their results show similar trends to ours.

We have also calculated the chemical potentials for Ne, Ar, Kr and Xe using the alchemical free energy method based on the He chemical potentials (see Methods). The calculated $D^{l-\text{Fe}/l-\text{MgSiO}_3}$ for Ne and Ar are compared with literature data in Fig. 1b, c. For Ne partition coefficients, we obtain $D^{l-\text{Fe}/l-\text{MgSiO}_3}$ of about $10^{-6}$ at both pressure and temperature conditions examined. This low

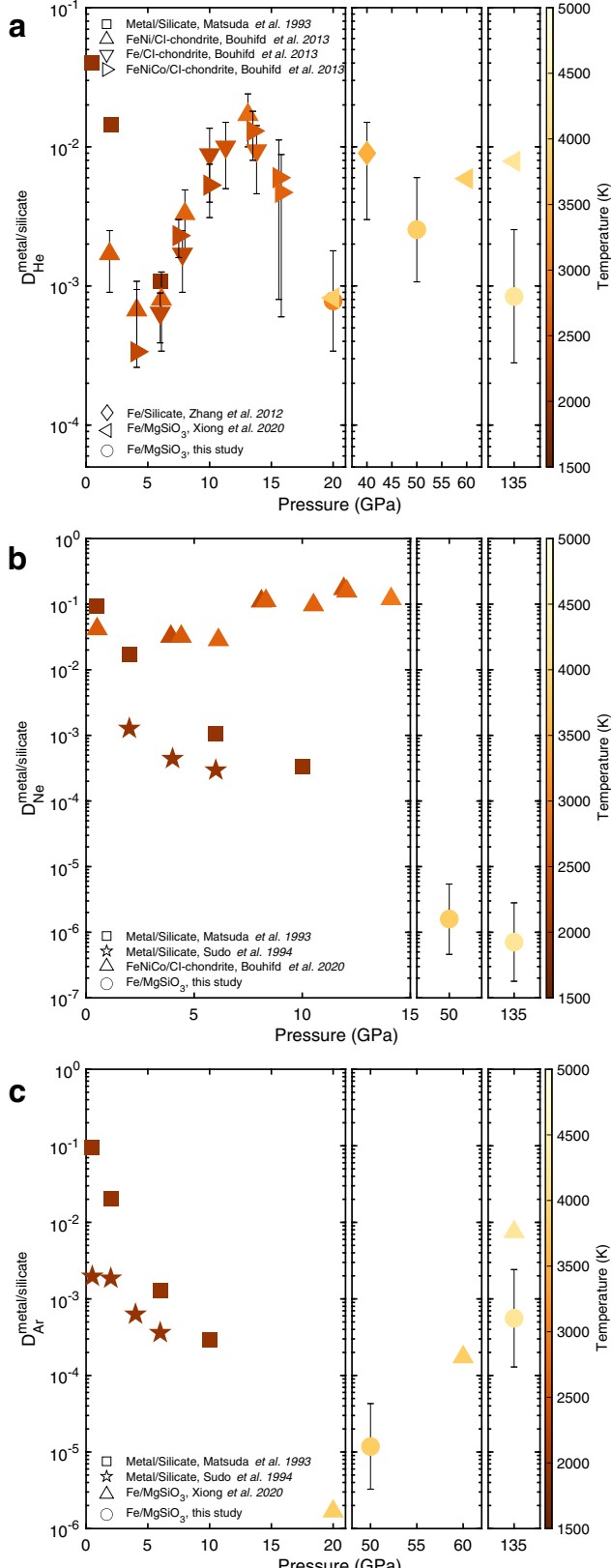

**Fig. 1 He, Ne and Ar partition coefficients.** Calculated and literature partition coefficients (concentration in weight) between liquid Fe and silicate melt at different pressures and temperatures for (**a**) He, (**b**) Ne and (**c**) Ar. **a** Helium shows some variation in $D$ with pressure depending on the particular study, however these changes are relatively small in comparison with the other noble gases (note different scale) and all results are consistent with a $D$ between $10^{-3}$ and $10^{-2}$. **b** There are some differences between different experimental studies for Ne at low pressure, with the more recent experiments of Bouhifd et al.[31] showing an almost constant $D$ across all pressures. In contrast our results agree better with the earlier experiments and we predict very low $D$s for Ne at high pressures. **c** Argon shows a decrease in $D$ until about 20 to 50 GPa followed by a strong increase. As discussed in the text, these trends can be explained by atomic size variations.

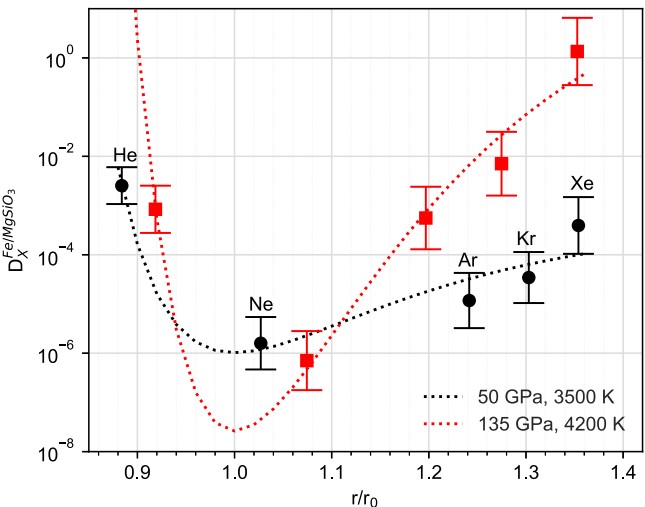

**Fig. 2 Partition coefficients of noble gases from calculations and the liquid-melt partition model.** Partition coefficients (concentration in weight) of noble gases between liquid Fe and silicate melt at 50 GPa, 3500 K and 135 GPa, 4200 K, plotted against the bond-length ratio of Mg–X to Mg–O (denoted as $r$ and $r_0$). Filled circles and squares are our calculated data from ab initio thermodynamics and the lines are from fitting the Lennard-Jones model. Fitted parameters can be found in Supplementary Information.

$D^{l-Fe/l-MgSiO_3} \sim 0.1$ at all pressures up to 14 GPa. However, as we show below, the low $D^{l-Fe/l-MgSiO_3}$ we obtain for Ne is consistent with the trend of partitioning for all the noble gases, where partitioning is controlled by the relative size of the noble gas to O in the silicate melt. Moreover, the $D^{l-Fe/l-MgSiO_3}$ for Ne of Bouhifd is an order of magnitude higher than He which means that planetesimal cores formed at low pressures would be enriched in Ne relative to He. However, the $^3$He/$^{22}$Ne ratio in iron meteorites shows no sign of fractionation of Ne from He and is close to the nebular ratio of ~1.5[32].

Our results for Ar are in reasonable agreement with those obtained by Xiong et al.[25] when considering the uncertainties (see Supplementary Information for more discussion). Taken together with the low-pressure experimental results, Ar partitioning decreases with pressure until about 50 GPa after which it increases. This is in contrast to He which exhibits the opposite behaviour, and with Ne which decreases across the whole pressure range. Kr and Xe also show an increase in partitioning at high pressure (see Fig. 2). The varying behaviour of the noble gases can be understood in terms of atomic size as is discussed in the next section.

value is consistent with the experiments of both Matsuda et al.[21] and Sudo et al.[30] which show a strong decrease in $D^{l-Fe/l-MgSiO_3}$ with increasing pressure. However, our results and the earlier experimental studies are at odds with the more recent study of Bouhifd et al.[31] which shows a remarkably constant

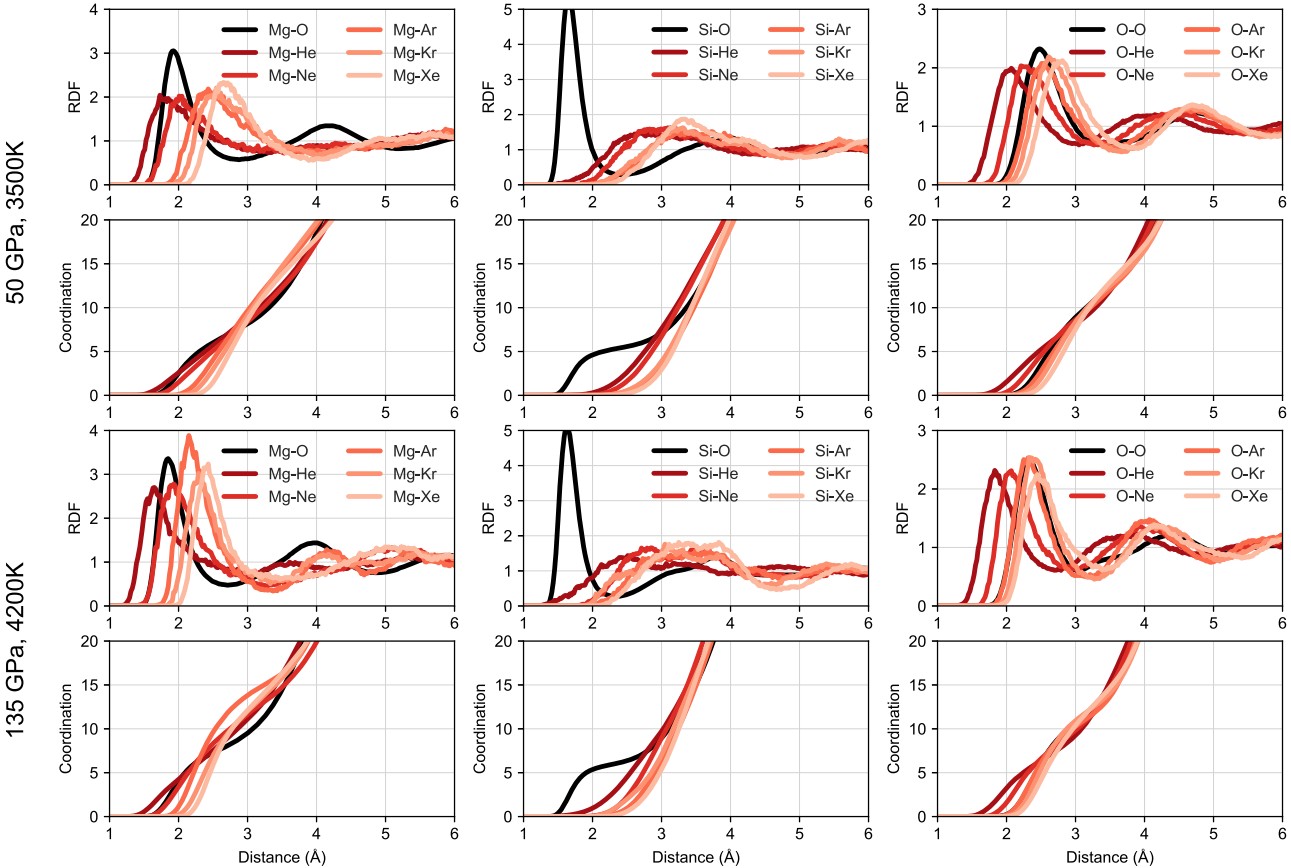

**Fig. 3 Radial distribution function and coordination numbers.** Results for silicate melts containing noble gases at 50 GPa, 3500 K and 135 GPa, 4200 K. For better comparison, the coordination numbers of noble gases with Mg, Si or O are scaled by the number of O atoms to the number of noble gas atoms.

**Liquid structure controls on iron-silicate melt partitioning.** Figure 2 plots our partition coefficients of He, Ne, Ar, Kr and Xe against the ratio of the Mg–X bond length to the Mg–O bond length (denoted as $r$ and $r_0$). At both pressures, $D^{l-Fe/l-MgSiO_3}$ exhibits a concave parabolic curve with bond-length radius, with a minimum around $r/r_0 = 1$. For mineral-melt partitioning, a plot of $\log D$ against cation radius is a convex parabola with the maxima of $\log D$ located at the crystal lattice site radius[33]. This is because the lattice mismatch strain in the solid is more pronounced than other factors and controls the partitioning. Although there is no such well-defined mismatch strain in liquids, the concave parabola plotted in Fig. 2 suggests that the partitioning of noble gases between liquid metal and silicate melt is controlled by the local environment in the silicate melt rather than the Fe-melt, and in particular the Mg–X bond length.

Further evidence that the silicate melt controls the partitioning can be seen in Fig. 3 which plots the radial distribution function (RDF) and coordination information for noble gas containing silicate melts. There are clear peaks in the noble gas-O and -Mg RDFs, which are at about the same distance as the first-neighbour distances of Mg–O and O–O. On the other hand, there are no Si-noble gas peaks at the first-neighbour distance of Si–O at all. This shows that noble gases prefer substituting for oxygen atoms that bond with Mg in the silicate melt rather than those that bond with Si, possibly because Mg–O bonding is weaker compared to Si–O bonding, and [MgOn] clusters more readily accept He[34,35]. In liquid Fe the structure is much simpler. Figure 4 shows the RDF and coordination information in liquid Fe, which indicate noble gases behave almost like Fe atoms in the closely packed liquid, except for He with a small volume that can also enter

interstitially. The first derivative of coordination peaks for Fe–Ne, Fe–Ar, Fe–Kr and Fe–Xe are similar to that of Fe–Fe (see Fig. 5), suggesting these pairs have well-defined bond lengths. However, there is no such a peak for Fe–He, again indicating that He is interstitial.

The 'bonding' of noble gases with Mg in silicate melt allows us to investigate the noble gas size effect on Mg–O bonding and subsequent chemical potentials. We find that the difference of noble gas chemical potentials between the liquid iron and silicate melt can be fitted to simple classic potentials like the Lennard-Jones model, i.e. $\bar{\mu}_X^{Fe} - \bar{\mu}_X^{MgSiO_3} = \bar{\mu}_X^{Fe} - \epsilon\left[\left(\frac{r_0}{r}\right)^{12} - 2\left(\frac{r_0}{r}\right)^6\right]$, where $\epsilon$ is the energy parameter and $r_0$ and $r$ denote the Mg–O and Mg–X bond lengths. Figure 2 also shows $D^{l-Fe/l-MgSiO_3}$ fitted to the Lennard-Jones model as a function of $r_0/r$, indicating such a model can successfully describe the calculated results.

The model also produces a tighter parabola at 135 GPa than at 50 GPa because the relative incompressibility of the silicate melt to the Fe melt is greater at higher pressure. As such the energy cost due to the atom size mismatch in the silicate melt is more costly at 135 GPa than at lower pressures, thus driving more of the larger noble gases into the Fe melt (see the discussion in Supplementary Information). Also, the potential energy surface (Fig. 6) becomes narrower and moves closer to the origin with increasing pressure. As a consequence, increasing pressure causes $\bar{\mu}_X^{MgSiO_3} - \bar{\mu}_X^{Fe}$ to decrease for He and Ne, while increasing for Ar, Kr and Xe. Thus, the partition coefficient increases significantly from 50 to 135 GPa for Ar, Kr and Xe, while decreasing slightly for He and Ne, thereby explaining the different pressure effect on the partitioning of light and heavy noble gases in Fig. 2.

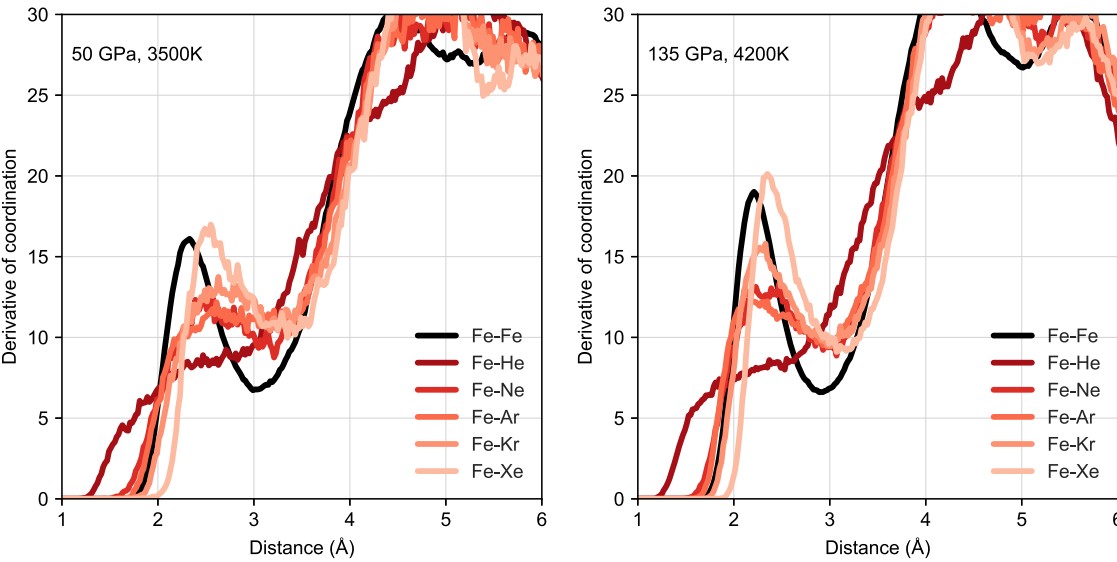

**Fig. 4 Radial distribution function and coordination numbers for noble gas containing liquid iron.** Calculation results for liquids at 50 GPa, 3500 K and 135 GPa, 4200 K. For better comparison, the coordination numbers of noble gases with Fe are scaled by the number of Fe atoms to the number of noble gas atoms.

**Fig. 5 Derivative of coordination numbers for noble gas containing liquid iron.** The first-order derivatives of Fe coordination numbers for coordination numbers shown in Fig. 4.

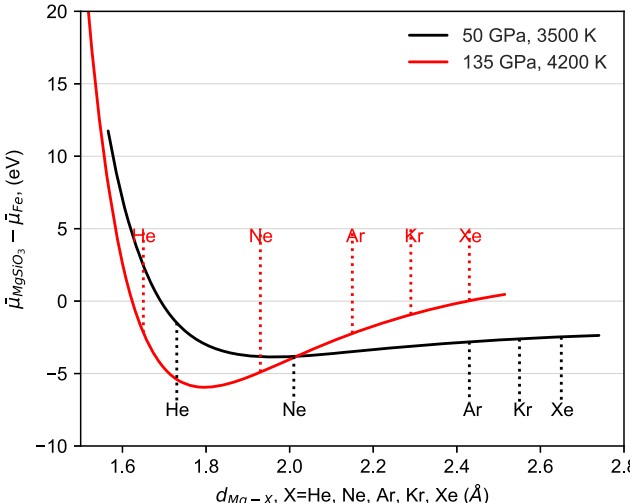

**Fig. 6 Fitted potential energy surface for $\bar{\mu}_X^{MgSiO_3} - \bar{\mu}_X^{Fe} = \epsilon[(\frac{r_0}{r})^{12} - 2(\frac{r_0}{r})^6] - \bar{\mu}_X^{Fe}$, X = He, Ne, Ar, Kr, Xe.** X 'substitutes' one O atom in [MgO$_n$] cluster and the size mismatch between O and X dominates the change in energy, while the chemical contribution is negligible.

**Primordial noble gases in the core**. In agreement with earlier studies, we find that the noble gases are lithophile under mantle core-forming conditions, and will partition strongly into the mantle silicate phase resulting in relatively low concentrations in the core. However, most of the original $^3$He in the magma ocean and early mantle has been degassed, and so the amount of He in the core could still be sufficient to supply OIBs with their primordial $^3$He/$^4$He signature as has been suggested before[20]. For instance, using the nebular ingassing model of Olson and Sharp[36] the magma ocean could have acquired 600 to 900 petagram ($10^{15}$ g or 1 Pg) of $^3$He (see Supplementary Information for details). Our partition coefficients at 50 GPa then predict the core to have about 0.75 Pg of $^3$He, similar to the abundance of $^3$He in the current mantle[37]. Thus a primordial signature of noble gases sourced from the core will not be completely masked by the mantle source.

However, our calculated partition coefficients for Ne are of the order of $10^{-6}$, three orders of magnitude lower than He. This extreme lithophile behaviour, driven by the fact that Ne has a similar atomic size to O in the silicate melt, limits Ne concentrations in the core to very low values unless the Earth accreted with orders of magnitude more Ne than He, for which there is no evidence. In contrast, available evidence suggests that He and Ne have similar solubilities in a magma ocean[38]. This difference in partitioning between He and Ne presents a key constraint on the core being the source of noble gases since our calculated partition coefficients would predict a very high $^3$He/$^{22}$Ne ratio, of the order of $10^3$ in the core. This ratio however, is three orders of magnitude higher than that observed in plume derived melts which have near solar values[1]. For instance, $^3$He/$^{22}$Ne in Galapagos and Iceland OIBs only reach 2–5[5]. Moreover, the observed trend between $^3$He/$^{22}$Ne and $^{20}$Ne/$^{22}$Ne in Icelandic OIBs[5] cannot be explained by mixing with a reservoir with such a high $^3$He/$^{22}$Ne of ~$10^3$.

At 50 GPa and 3500 K, the partition coefficients for the other three noble gases, Ar, Kr and Xe are between those of He and Ne, although $D$ for the heaviest noble gas, Xe, is near to that of He. These heavy gases do not have a clear accretionary core signal and since their concentrations are heavily overprinted by atmospheric recycling into the mantle[39], they are not normally used to study core uptake and release. Interestingly, the partition coefficients for

these three species increase substantially with pressure by two to three orders of magnitude. Depending on their concentration in the lower most mantle now, the core may be undersaturated in the heavy noble gases, opening up the possibility that they now diffuse into the core rather than the other way around.

**Primordial reservoir in the mantle**. Our results on He and Ne show therefore, that the often cited view of core-formation—whereby Fe melts separate and sink from a (mostly) molten silicate Earth—cannot supply noble gases to the core in the ratios observed in OIBs. This then leaves us with two options. One is that the core is simply not the reservoir of primordial noble gases seen in OIBs. This then requires the source of primordial noble gases to be in the mantle, with either LLSVPs, BEAMs, or ULVZs as likely candidate reservoirs[40]. Alternatively, the core has inherited its noble gases from the unequilibrated cores of planetesimals[41]. Using low-pressure and temperature He partitioning experiments, Roth et al.[41] showed that He behaves as a moderately siderophile element when partitioning between silicate minerals (olivine) and an FeS melt. Since He is an incompatible element (i.e. it partitions strongly into silicate melts rather than mineral phases), its partition coefficient between liquid iron and solid silicate minerals is much higher than between liquid iron and silicate melt. If the first thing to melt in a planetesimal are the FeS phases, then Roth et al. hypothesise that planetesimal cores which formed at low pressures before the planetesimal melted, could have incorporated large amounts of noble gases—much more than from core separation from a magma ocean—and so only a few percent of these planetesimal cores are needed to dominate the noble gas budget of the Earth's core.

However, for planetesimal-derived noble gases in the Earth's core to explain the solar ratios of noble gases seen in OIBs requires that (1) there is no fractionation of Ne from He during planetesimal core formation, and (2) that they also remain unfractionated during the path the gases take from the core to the surface. The first of these requirements is supported by observations from some iron meteorites showing that planetesimal cores have approximately solar-like noble gas ratios[32], suggesting little fractionation of He and Ne at low pressures. This is also consistent with similar low-pressure partitioning data of He and Ne shown in Fig. 1, albeit between Fe-melt and silicate-melt. However, the second requirement is unlikely to be met. Our results show a large effect of pressure on the partitioning of Ne relative to He, suggesting that equipartitioning of Ne and He between the core and mantle is unlikely. In other words, a core with solar-like ratios of He and Ne should be hugely out of equilibrium with the ratio in the mantle, and since diffusion drives material towards equilibrium, this will strongly affect the diffusion rates of Ne and He into the mantle and likely fractionate them.

There are other arguments against the core being the source of OIB noble gases. Firstly bulk entrainment of core material would be accompanied by a fingerprint of highly siderophile elements (HSEs)[42], and even a small contribution from the core would imprint the rising plumes with a distinguishable HSEs signal[43]. However, HSEs abundances in OIBs are normal: there is no resolvable signal suggesting they are from HSE-enriched sources[42]. The alternative to bulk incorporation of core material is diffusion of He and Ne out of the core, perhaps using LLSVPs or ULVZs as a staging post to concentrate them in the plume source. However, the diffusion coefficients of Ne and He must be very similar in order to not fractionate them, and moreover, their very different partition coefficients means that even a trace of melt would fractionate them.

None of these arguments prove that the core cannot be the source of He and Ne in OIBs. In contrast, our results show that the core

cannot have the $^3$He/$^{22}$Ne ratios seen in OIBs and so it would remarkably fortunate if they were somehow fractionated back to solar-like ratios as they made their way to the source of hotspots and then to OIBs. Moreover, even if the core did have solar-like noble gas ratios inherited from a small fraction of undifferentiated planetesimal cores, given the enormous difference in partitioning predicted by this work and that of Roth et al.[41] it would also be a huge coincidence if these ratios are maintained all the way to OIBs. We conclude, therefore, that the source of these noble gases lies in somewhere in the mantle and not in the core.

A well isolated primordial reservoir in the lower mantle does not contradict with the correlation between high $^3$He/$^4$He and other isotope data in OIBs. For instance, although there is an observed correlation between $^{182}$W and $^3$He/$^4$He[40] in different hotspots, this is best explained by mixing with core derived $^{182}$W (perhaps ULVZs) and a mantle side reservoir with high $^3$He/$^4$He but low $^{182}$W (perhaps LLSVPs)[40]. It has also been suggested that the core may have inherited most of the Earth's iodine and thus should have a high $^{129}$Xe/$^{130}$Xe ratio[44], however, since the mantle's Xe is so dominated by recycled Xe, identifying a deep Earth origin for Xe is difficult[45]. In fact any correlation between putative core material and high $^3$He/$^4$He does not in itself distinguish between a core or deep mantle source of noble gases. Indeed, in this scenario it would be surprising if any core addition to mantle plumes did not entrain material from the deep mantle, including the primordial gases.

In conclusion, while it is possible that the core may contain significant amounts of some noble gases, it can only have a very low concentration of Ne. This precludes the core from being the source of noble gases seen in OIBs and so there must be a mantle reservoir of noble gases which has remained unmixed with the rest of the mantle for 4.6 Gyrs.

## Methods

**Chemical potential calculations**. We performed thermodynamic integration to obtain free energies for noble gas containing liquid iron and silicate melt, from which the chemical potentials of noble gases can be obtained. We first calculated the chemical potentials of He in both liquid iron and silicate melt. We took the ideal gas as the reference system in free energy calculations. Therefore, we decomposed the Gibbs free energy $G(p, T, x)$ under specific pressure ($p$), temperature ($T$) and solute concentration ($x$) into two terms: the ideal gas mixing entropy term $TS_{mix}$ ($T$ is the temperature and $S_{mix}$ is the mixing entropy) and all the left $\bar{G}(p, T, x)$[46].

Rather than directly referencing the ab initio system to the ideal gas, we made use of the truncated repulsive Lennard-Jones system, namely the Weeks-Chandler-Andersen (WCA) gas system[47]. The free energy of the WCA system referenced to the ideal gas is known[48,49]. This reduces the computation load. The WCA potential $\phi_{WCA}$ is

$$\phi_{WCA}(r) = \begin{cases} 4\varepsilon\left[\left(\frac{\sigma}{r}\right)^{12} - \left(\frac{\sigma}{r}\right)^6\right] + \varepsilon, & r \leq 2^{1/6}\sigma \\ 0, & r > 2^{1/6}\sigma \end{cases} \quad (1)$$

where $\varepsilon$ and $\sigma$ are the energy parameter and length parameter, respectively. We do thermodynamic integration to reference the ab initio system to the WCA system. Therefore, Gibbs free energy is expressed as

$$G(V) = P_0 V + F(V) = P_0 V + F_{WCA} + \frac{1}{N}\int_0^1 \langle\phi\rangle_\lambda d\lambda \quad (2)$$

where $V$ is the volume per atom under the target pressure $P_0$, and $N$ is the number of atoms in the system. $F_{WCA}$ is the known free energy of the WCA gas[48,49]. $\lambda$ is the coupling parameter, and the potential function is $\phi_\lambda = \phi_{WCA} + \lambda\phi$, where $\phi$ and $\phi_{WCA}$ are the ab initio and WCA potential functions, respectively. For the integration, we used a six-point Gaussian-Legendre quadrature that is sufficient to converge the integral[47]. $\sigma$ was chosen to make the cutoff distance ($2^{1/6}\sigma$) shorter than the minimal bond length in the ab initio system. $F_{WCA} = F_{ig} + F'_{WCA}$, where $F_{ig}$ is the idea gas free energy and $F'_{WCA}$ is the extra free energy part of the WCA system. $F_{ig}$ for the multicomponent system is

$$F_{ig} = -k_B T \sum_{i=1}^n \ln Z_i \approx k_B T \sum_{i=1}^n N_i\left[\ln\left(\frac{\Lambda_i^3}{V}\right) - 1 + \ln\left(\frac{N_i}{N}\right)\right] \quad (3)$$

We then obtained a series of $G(p, T, x)$ as a function of He concentration for liquid Fe–He mixture and silicate melt-He mixture. Subsequently, the He chemical potential $\mu(p, T, x)$ can be obtained from derivatives of $G$ and is also expressed by two terms: the partial ideal gas mixing entropy term $TS_{mix}^{He}$ and the pure component term $\bar{\mu}(p, T, x)$. We have

$$\mu(p, T, x) = \bar{\mu}(p, T, x) - TS_{mix}^{He} \quad (4)$$

where $S_{mix}^{He} = -k_B \ln x$, for He in the $Fe_{1-x}He_x$ mixture. $k_B$ is the Boltzmann constant. The mixing entropy for the $(MgSiO_3)_{1-x}He_x$ mixture is

$$S_{mix}^{(MgSiO_3)_{1-x}He_x} = -k_B\left(x\ln\left(\frac{x}{5-4x}\right) + (1-x)\left(2\ln\left(\frac{1-x}{5-4x}\right) + 3\ln\left(\frac{3-3x}{5-4x}\right)\right)\right) \quad (5)$$

And for He in the $(MgSiO_3)_{1-x}He_x$ mixture,

$$S_{mix}^{He} = S_{mix}^{(MgSiO_3)_{1-x}He_x} + (1-x)\partial S_{mix}^{(MgSiO_3)_{1-x}He_x}/\partial x \quad (6)$$

We then calculated the chemical potentials of Ne, Ar, Kr and Xe by using the alchemical free energy method. This method has been well documented in previous publications[50,51]. We converted one He atom in the liquid iron or silicate melt into X (Ne, Ar, Kr or Xe) atom, and the difference of chemical potential between He and X can be obtained from the change of free energy. Since $\mu_{He}$ is already known, the chemical potentials of Ne, Ar, Kr and Xe can be obtained. In practice we first obtained the free energy difference ($\triangle F$) for the system under constant volume and constant temperature, and then a correction to the constant pressure can be made conveniently[50]. At constant volume, the difference of Helmholtz free energy is

$$\triangle F = F(N_{He} - 1, N_X + 1) - F(N_{He}, N_X)$$
$$= -k_B T\ln\left\{\frac{N_{He}\Lambda_{He}^3}{(N_X + 1)\Lambda_X^3} \times \frac{\int_V d\boldsymbol{R}\exp[-U(N_{He}-1, N_X+1)/k_B T]}{\int_V d\boldsymbol{R}\exp[-U(N_{He}, N_X)/k_B T]}\right\} \quad (7)$$

where $F(N_{He}, N_X)$ is the free energy of the system (liquid iron or silicate melt) with numbers of $N_{He}$ He atoms and $N_X$ X atoms, $F(N_{He} - 1, N_X + 1)$ is for the system with $(N_{He} - 1)$ He atoms and $(N_X + 1)$ X atoms. $k_B$ is the Boltzmann constant and T is the temperature. $U(N_{He}, N_X)$ dependent on the coordinates of all atoms $\boldsymbol{R}$ is the potential energy for the system with $N_{He}$ He atoms and $N_X$ X atoms. $U(N_{He} - 1, N_X + 1)$ is the potential energy for the system with $(N_{He} - 1)$ He atoms and $(N_X + 1)$ X atoms. $\Lambda_{He}$ and $\Lambda_X$ denote the thermal wavelengths of He and X.

Further technique details and density functional calculation details were provided in the Supplementary Information.

## Data availability

The raw outputs can be accessed in the UK National Geoscience Data Centre (NGDC) (https://doi.org/10.5285/8c594384-9503-4902-b81b-9e5f643f2328). Any additional data can be requested by e-mailing the corresponding author.

## Code availability

The Vienna Ab Initio Simulation Package (VASP) is a proprietary software available for purchase at https://www.vasp.at/.

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

## Acknowledgements

We acknowledge the support of NERC (grant NE/S01134X/1). Y.L. also thanks the support from CAS Hundred Talents Programme and NSFC (grant 42173040). We acknowledge use of the NEXCS system, a collaborative facility supplied under the Joint Weather and Climate Research Programme, a strategic partnership between the Met Office and the Natural Environment Research Council. This work also used the ARCHER UK National Supercomputing Service and the supercomputing system in the Supercomputing Centre of University of Science and Technology of China.

## Author contributions

Y.L. carried out the simulations and analysis. L.V. and J.B. supervised the project. Y.L., L.V., J.B. and C.B. all contributed to data analysis and writing the paper.

## Competing interests

The authors declare no competing interests.
