## [Peer Review File · Nature Communications]

Primitive noble gases sampled from ocean island basalts cannot be from the Earth's coreEditorial Note: This manuscript has been previously reviewed at another journal that is not operating a transparent peer review scheme. This document only contains reviewer comments and rebuttal letters for versions considered at *Nature Communications*.

REVIEWER COMMENTS

Reviewer #1 (Remarks to the Author):

The authors present new ab initio calculations of noble gas partition at core/mantle boundary temperatures and pressures. They conclude that the core is not the source of light noble gases in OIBs and that they must be derived from an isolated deep source, perhaps one of the deep mantle reservoirs LLSVPs, ULVZs, BEAMS or an, as yet, unidentified reservoir. The manuscript clearly presents the output of new modelling and compares the outcome with existing experimental data at lower pressures. The modelling corroborates earlier less sophisticated calculations but crucially shows that He partition between silicate and Fe-rich liquids is very different to Ne and that it is possible to put He into the core. However, the consequence of the difference is that while the core might contain sufficient ^3He , it could not be the source of He and Ne in OIBs.

The manuscript is much improved from an earlier version. The discussion of ab initio calculations for the heavier noble gases is now coherent and develops some useful themes. The authors note that their calculations show heavy noble gases have partition coefficients similar to He and thus have the potential to partition into the core. However, primitive heavy noble gases are masked by much higher concentrations recycling from the Earth's surface via subduction. Further, the authors speculate that high surface derived heavy noble gases are more likely to partition into an undersaturated than be released. Thus the calculations may open new areas of noble gas geochemistry research, considering the various reservoir interactions, although there is still no clear candidate reservoir that satisfies all the evidence.

To summarise, the work is of great interest and value. It is an important step towards uncovering the structure and evolution of the deepest Earth and its reservoirs.

Reviewer #2 (Remarks to the Author):

This article that uses ab initio calculations to determine the partitioning of noble gases between Earth's Fe-rich core and silicate mantle. The primary finding is that partitioning at high pressure should result in huge fractionations between helium and neon, which is not observed in mantle-derived rocks. This, in turn, is used to argue against a core origin for the high $^3\text{He}/^4\text{He}$ ratio measured in some ocean islands basalts. If true, these findings could inform our understanding of reservoirs and dynamics in the deep mantle.

Comments/suggestions:

1) There are many studies discussing the pressure of core formation. While those quoted values may change, they are usually discussed as an "average" pressure of core formation, but the paper here appears to base their conclusions on a maximum pressure of formation. I would like to see the process of core formation discussed in a bit more detail especially as it pertains to the partitioning of the noble gases at specific pressures.

2) I cannot speak to the details of the simulations presented here but it would have been nice to have seen whether this method could reproduce published values at lower pressures. This is particularly important as some of the extreme partitioning of the noble gases in this current study appear at a pressure where little to no data is available. For example, for He there is a huge change in partitioning between 20 and 40 GPa – my first question is whether the technique here reproduces the values at 20GPa? Similarly, for Ne that is a jump in partitioning from 15 to 50 GPa.

3) There is a complementary paper published Wang et al. (2022) Partitioning of noble gases (He, Ne, Ar, Kr, Xe) during Earth's core segregation: A possible core reservoir for primordial noble gases that needs to be discussed here. Its omission is not the author's fault as Wang et al. was most likely published after this paper was submitted. But the results of Wang et al. appear to be at odds with this paper in that the predicted He/Ne ratio at high pressure is orders of magnitude smaller.

4) If these discrepancies can be addressed, and an extremely high He/Ne is warranted, I would appreciate a deeper discussion of the impact. Currently, they discuss how it may be in one reservoir or another, but what is the impact of being in one reservoir versus another? What does it tell us about the formation of that reservoir? What does it tell us about dynamics of the lower mantle? For example, the authors start their manuscript discussing how $^3\text{He}/^4\text{He}$ ratios are correlated with $^{20}\text{Ne}/^{22}\text{Ne}$ – how does that fit into the context of their results? There have also been several papers suggesting $^3\text{He}/^4\text{He}$ ratios are correlated with ^{182}W values (Mundl et al.; Rizo et al.). If the noble gases are not coming from the core, then does that mean ^{182}W values are not coming from the core consistent with some ^{142}Nd values reported recently (Horan et al. 2018)? What does it mean with regards to the I/Xe system? The authors looked at Xe partitioning – is there anything to say there?

Response to Reviewers' Comments:

Reviewer 2:

R2.1) Comment: *There are many studies discussing the pressure of core formation. While those quoted values may change, they are usually discussed as an “average” pressure of core formation, but the paper here appears to base their conclusions on a maximum pressure of formation. I would like to see the process of core formation discussed in a bit more detail especially as it pertains to the partitioning of the noble gases at specific pressures.*

Response: Sorry, we did not mean to suggest everything was based on a maximum pressure of core formation, but that the low-pressure data are too low to be used robustly to estimate core concentrations under core-forming conditions. We have now expanded our discussion accordingly.

We have now added the following text:

“Core-mantle differentiation is highly likely to be a polybaric process as metals droplets will re-equilibrate progressively on the way to the core. Indeed, Rubie et al.²³ modelled the Earth’s accretion and differentiation by considering the partitioning of Fe, Si, O, Ni, Co, W, Nb, V, Ta and Cr, and found an average equilibration at pressures larger than 50 GPa is required for core formation. This average high-pressure differentiation is also consistent with the highly siderophile elements abundances in the mantle²⁴. Using the existing low-pressure data to constrain the partitioning of the core is, therefore, likely to have large uncertainties.”

R2.2) Comment: *I cannot speak to the details of the simulations presented here but it would have been nice to have seen whether this method could reproduce published values at lower pressures. This is particularly important as some of the extreme partitioning of the noble gases in this current study appear at a pressure where little to no data is available. For example, for He there is a huge change in partitioning between 20 and 40 GPa – my first question is whether the technique here reproduces the values at 20GPa? Similarly, for Ne that is a jump in partitioning from 15 to 50 GPa.*

Response: Ab initio thermodynamics methods are well-established tools. The results are reliable if physical considerations are complete and calculations are sound. The approach used here has been used recently to determine the water partition between liquid iron and silicate melt. The ab initio results agree well with experimental data at low pressures. Previously we have checked the accuracy of chemical potentials calculated in this study by converting noble gases to H at 50 GPa, and the resulted H chemical potentials agree well with published results.

In order to compare to low-pressure experimental data in this study, we further calculated the He chemical potentials at 20 GPa. The partition coefficient has been plotted in Fig. 1. It can be seen that the calculated D lies within the experimental uncertainty of those at lower pressures and seems to follow the trend. However, whether the ups and downs shown in the experimental data are robust is not clear (i.e. the very low pressure results of Matsuda are not seen in the later experiments),

however, what is clear is that D for He is between about 10^2 and 10^3 across all mantle conditions.

We did not calculate D for Ne at 20 GPa. First, the three sets of experiments differ so large and so any match with one of them will not be particularly convincing. Second, the core-mantle differentiation occurs at pressures ~ 50 GPa and higher.

Changed texts:

“As can be seen, our 20 GPa partition coefficient lies within the range of experimental data and agrees with the calculation by Xiong et al.²⁵. Our 50 GPa partition coefficient is in good agreement with the highest pressure experimental data and the calculations of Zhang et al.²²”

“Besides, we did alchemical free energy calculation at 20 GPa and 50 GPa in $Fe_{64}H_2$, and obtained the chemical potential differences between H and He, and by using the reported H chemical potentials⁴ we obtained another He chemical potentials. The new chemical potentials agree well with the ones above.”

Changed Fig. 1a:

R2.3) Comment: *There is a complementary paper published Wang et al. (2022) Partitioning of noble gases (He, Ne, Ar, Kr, Xe) during Earth’s core segregation: A possible core reservoir for primordial noble gases that needs to be discussed here. Its omission is not the author’s fault as Wang et al. was most likely published after this paper was submitted. But the results of Wang et al. appear to be at odds with this paper in that the predicted He/Ne ratio at high pressure is orders of magnitude smaller.*

Response: It is unfortunate that this has been published, but in fact (1) their results have much larger uncertainties than ours and secondly the way they obtain D from dG is not correct.

(For a detailed comparison of the methods and results, please go to Supplementary Information.)

Taking the uncertainty first, Wang *et al.* used the same method as Xiong *et al.* to derive the chemical potentials. We have explained previously in the text that their method is likely to have a substantial uncertainty, because their chemical potentials were calculated based on the free energies using only **one** concentration of noble gases in each phase rather than the five or six used here. The chemical potential is the derivative of free energy with respect to composition, so using only one concentration of noble gas to obtain the chemical potential will lead to a substantial uncertainty in D (~ 3 orders of magnitude), that is much greater than ours.

Secondly, in order to obtain D for a trace element you need to equate the chemical potentials of the trace element in each phase. This is shown clearly in the chapter by Bennet in 2nd edition of the Encyclopedia of Geology. This is exactly what we have done. Instead, both Wang *et al.* and Xiong *et al.* use their dG to obtain a K_d . First of all they both define a different K_d (neither of which gives a D), but secondly, when using $K_d = \exp(-dG/kT)$, they should use dG^0 , the difference in free energies of the pure substance (again see the chapter by Bennet). However, both these studies use dG – the difference in the full G which include the mixing terms. This is wrong and results in a temperature dependence of the partitioning which is far too high.

Nonetheless, we find the chemical potentials from Wang *et al.* actually show a very similar trend to ours. We plotted their chemical potentials at 40, 60 and 135 GPa in Fig.S6, in comparison with our results. The chemical potential difference of Ne between liquid iron and in silicate melt is much higher than any of the other noble gases. This occurs at all pressure, from both this study and Wang *et al.*. If we use the chemical potentials calculated by Wang *et al.*, the partition coefficients should be close to ours and the core He/Ne ratio should also be close to 1000, as can be seen from Fig.S7.

We also tried to reproduce their reported partition coefficients from their reported chemical potentials, but failed. We attribute this to a mistake in their derivation of partition coefficients – Equations 1 to 5 in their paper. For example, $\Delta\mu$ for Ar at 20 GPa is 3.84 eV from Xiong *et al.* and 4.92 eV from Wang *et al.*, so D from Wang *et al.* should be much lower than Xiong *et al.* However, the opposite is reported and the D of 5.29×10^{-6} at 2500 K from Wang *et al.* is much higher than Xiong *et al.*, about 1×10^{-7} at 2500 K. The large difference in the chemical potentials between Wang *et al.* and Xiong *et al.* also corroborates our claim about the substantial uncertainties inherent in using only one concentration of noble gas in their approach.

We thus cannot fully discuss the results of Wang *et al.* in the text, but we have compared our chemical potential calculations with their results and described this in detail in the Supplementary Information.

Fig. S6. The chemical potential difference between in liquid iron and silicate melt for noble gases.

Fig. S7. Comparison between our partition coefficients and those of Wang et al. 2022.

changed texts:

“We have not considered the newer results of Wang et al since they are also based on an incorrect use of the free energy in obtaining Ds. As we show in the Supplementary Information, when using their Gs to equate chemical potentials, their results show similar trends to ours.”

*Supplementary Information -- Text S2. Comparison with other theoretical results
Supplementary Information -- Figure S4: Comparison between our He partition*

coefficients with literature partition coefficients (in weight) between liquid Fe and silicate melt.

*Supplementary Information -- **Figure S5:** Comparison between our Ar partition coefficients with literature partition coefficients (in weight) between liquid Fe and silicate melt.*

*Supplementary Information -- **Figure S6:** The chemical potential difference between in liquid Fe and in silicate melt.*

*Supplementary Information -- **Figure S7:** Comparison between our partition coefficients and those of Wang et al. 2022.*

*Supplementary Information -- **Table S1:** Chemical potential ($\bar{\mu}$) difference between in liquid iron and in silicate melt (in eV/atom).*

R2.4) Comment: *If these discrepancies can be addressed, and an extremely high He/Ne is warranted, I would appreciate a deeper discussion of the impact. Currently, they discuss how it may be in one reservoir are another, but what is the impact of being in one reservoir versus another? What does it tell us about the formation of that reservoir? What does it tell us about dynamics of the lower mantle? For example, the authors start their manuscript discussing how $^3\text{He}/^4\text{He}$ ratios are correlated with $^{20}\text{Ne}/^{22}\text{Ne}$ – how does that fit into the context of their results? There have also been several papers suggesting $^3\text{He}/^4\text{He}$ ratios are correlated with ^{182}W values (Mundl et al.; Rizo et al.). If the noble gases are not coming from the core, then does that mean ^{182}W values are not coming from the core consistent with some ^{142}Nd values reported recently (Horan et al. 2018)? What does it mean with regards to the I/Xe system? The authors looked at Xe partitioning – is there anything to say there?*

Response: Thanks for the suggestion, and although we agree that building on our finding might provide other interesting and important findings, we also feel that the Ne/He system stands on its own in showing that the core cannot be the origin of the main noble gas signature seen in OIBs – that of 3/4 He and Ne. Nevertheless, we have addressed the ^{182}W systematics as suggested by the reviewer. In fact this agrees with our findings. I/Xe we feel is a much harder system to say much about since the mantle Xe is dominated by recycled Xe and indeed a deep Earth Xe signature may not actually exist at all (Bekaert et al 2019). Hopefully the following text at the end of the paper is a balanced and pragmatic end to the paper.

changed texts:

“A well isolated primordial reservoir in the lower mantle does not contradict with the correlation between high $^3\text{He}/^4\text{He}$ and other isotope data in OIBs. For instance, although there is an observed correlation between ^{182}W and $^3\text{He}/^4\text{He}$ ³⁷ in different hotspots, this is best explained by mixing with core derived ^{182}W (perhaps ULVZs) and a mantle side reservoir with high $^3\text{He}/^4\text{He}$ but low ^{182}W (perhaps LLSVPs)³⁷. It has also been suggested that the core may have inherited most of the Earth's iodine and thus should have a high $^{129}\text{Xe}/^{130}\text{Xe}$ ratio⁴¹, however, since the mantle's Xe is so dominated by recycled Xe, identifying a deep Earth origin for Xe is difficult⁴². In fact any correlation between putative core material and high $^3\text{He}/^4\text{He}$ does not in itself distinguish between a core or deep mantle source of noble gases. Indeed, in this scenario it would be surprising if any core addition to mantle plumes did not entrain material from the deep mantle, including the primordial gases.

In conclusion, while it is possible that the core may contain significant amounts of

some noble gases, it can only have a very low concentration of Ne. This precludes the core from being the source of noble gases seen in OIBs and so there must be a mantle reservoir of noble gases which has remained unmixed with the rest of the mantle for 4.6 Byrs. ”

REVIEWERS' COMMENTS:

Reviewer #2 (Remarks to the Author):

I am satisfied and would like to thank the authors for taking to the time to address my questions.